# Synchronization of spin-driven limit cycle oscillators optically levitated in vacuum

Oto Brzobohatý [1] ✉, Martin Duchaň[1], Petr Jákl [1], Jan Ježek[1], Martin Šiler [1], Pavel Zemánek[1] & Stephen H. Simpson [1] ✉

We explore, experimentally and theoretically, the emergence of coherent coupled oscillations and synchronization between a pair of non-Hermitian, stochastic, opto-mechanical oscillators, levitated in vacuum. Each oscillator consists of a polystyrene microsphere trapped in a circularly polarized, counter-propagating Gaussian laser beam. Non-conservative, azimuthal forces, deriving from inhomogeneous optical spin, push the micro-particles out of thermodynamic equilibrium. For modest optical powers each particle shows a tendency towards orbital circulation. Initially, their stochastic motion is weakly correlated. As the power is increased, the tendency towards orbital circulation strengthens and the motion of the particles becomes highly correlated. Eventually, centripetal forces overcome optical gradient forces and the oscillators undergo a collective Hopf bifurcation. For laser powers exceeding this threshold, a pair of limit cycles appear, which synchronize due to weak optical and hydrodynamic interactions. In principle, arrays of such Non-Hermitian elements can be arranged, paving the way for opto-mechanical topological materials or, possibly, classical time crystals. In addition, the preparation of synchronized states in levitated optomechanics could lead to new and robust sensors or alternative routes to the entanglement of macroscopic objects.

Over the preceding decade, levitational optomechanics has emerged as a versatile platform for addressing crucial questions in the physical sciences, ranging from the macroscopic limits of quantum mechanics[1] to the thermodynamic limits of computation[2]. It makes use of optical forces, which are generated when light scatters from small particles. These forces can confine or suspend isolated particles in vacuum, or induce structured interactions, known as *optical binding forces*, amongst collections of them[3]. Supplementing optical with electrostatic forces[4], and combining with an optical cavity[5] results in a reconfigurable experimental system with widely tunable reactive and dissipative forces, capable of supporting dynamical effects across multiple physical regimes[6].

Most significantly, these techniques have recently enabled motional cooling of nanoparticles towards and into their quantum mechanical ground state, in single and multiple degrees of freedom[7–10] with the future promise of macroscopic entanglement[11–13].

In the classical domain, the harmonic potentials associated with optical tweezers can confine cooled particles forming high Q oscillators with exquisite force sensitivity[14,15]. However, optomechanical systems can exhibit far richer behaviour. Optical forces, including interaction forces, are, in general, non-conservative[16–19], and can be holographically sculpted[20,21]. In combination, arbitrarily structured non-conservative and non-linear forces, thermal fluctuations and dissipation provide the necessary ingredients for numerous stochastic and dynamic phenomena which are not only of intrinsic interest, but could also have novel applications in sensing and metrology. Examples include autonomous stochastic resonance[22], coherence resonance[23],

[1]The Czech Academy of Sciences, Institute of Scientific Instruments, Královopolská 147, 612 64 Brno, Czech Republic. ✉e-mail: otobrzo@isibrno.cz; simpson@isibrno.cz

stochastic bifurcations[24] and stochastic synchronization, all of which are exploited by nature in the sense apparatus of animals[25,26].

Synchronization of noisy, limit cycle oscillators is an archetypal non-equilibrium effect[27,28]. For mesoscopic systems the phenomenon has been studied extensively in the low Reynolds number (low Re) regime, see[29]. The significance of these low Re systems lies in their application to micro-biology, and on their reliance on coupled dissipative forces (i.e. hydrodynamic coupling) to achieve synchronization. The system we will study here is, in many respects, fundamentally different: (i) our system is underdamped, with steady state conditions formed by a delicate balance between (reactive) optical, (dissipative) hydrodynamic and inertial forces, (ii) coupled reactive forces play a key role in synchronization, and (iii) our system is completely unconstrained, spontaneous and autonomous, in comparison with model low Re systems, in which both the paths followed by the particles, and the force profiles that drive them, are prescribed by the experimenter[30,31]. This final point is important. In contrast, the stochastic trajectories followed by the particles in our system derive from underlying physical principles, and so can be used to test emerging concepts in stochastic thermodynamics such as thermodynamic uncertainty relations[32]. These issues are discussed further in Supplementary Note (VII).

The ability to form coherent, coordinated non-equilibrium states in the underdamped, mesoscopic regime, such as the synchronized states we will describe here, could have applications in sensor arrays[33], suppressing phase noise and natural variations in fundamental frequencies[34,35]. These linearly non-conservative oscillators are particular examples of a broader class of non-Hermitian oscillator[36,37], characterised by broken time reversal symmetry and a capacity to exchange energy with the environment. Arrays of such non-Hermitian units can form topological phases and exhibit exponential sensitivity through the well know skin effect[33,38–40].

These phenomena have been successfully realized in the classical domain with the use of micro-robotics[41,42]. Optical forces, such as those considered here, offer a route to realize similar effects, spontaneously, in the mesoscopic regime. Moreover, under appropriate conditions, such systems may also act like classical time crystals[43,44]. Developing appropriate cooling techniques (see refs. 45,46) could take these effects towards the quantum regime and provide experimental access to mesoscopic quantum dynamic phenomena such as quantum synchronization or entanglement[13,47,48]. This issue is revisted in the Discussion and in Supplementary Note (VIII). In this article we provide a demonstration of the synchronization of a pair of optomechanical limit cycle oscillators, driven by inhomogeneous optical spin. As described above, this is a key step in realising a number of applications, ranging from sensor arrays to novel topological oscillators. Our limit cycle oscillators comprise polystyrene microspheres trapped in circularly polarized, counter-propagating optical beams, in vacuum. Each oscillator is linearly non-conservative, due to azimuthal components of momentum associated with inhomogeneous optical spin[49–51], and coupled through weak hydrodynamic and optical interactions. Analogous effects can be induced via birefringence[52], or phase difference[53,54]. Below a critical, threshold power, we observe biased Brownian motion, featuring correlations which strengthen with increasing optical power. At threshold, a bifurcation occurs[55] and the stable trapping points are replaced with noisy limit cycles (i.e. stable, self-sustained, periodic motions) that form robust synchronized states with characteristic detuning behaviour[23].

## Results

### Qualitative experimental observations

We explore the emergent, coordinated motion of a pair of non-conservative optomechanical oscillators, see Fig. 1a for the experimental geometry. Each oscillator consists of a polystyrene microsphere (nominal radius, $a = 425$ nm) confined within counter-

propagating, circularly polarized (CPCP) Gaussian beams (wavelength $\lambda = 1064$ nm and beam waist 900 nm) with ambient pressure is 17 mbar, equivalent to an effective viscosity $\mu \approx 1.15 \, \mu Pa \, s$[56]. The axes of the CPCP beams are parallel and separated by a distance $d$. Circular polarization gives rise to azimuthal components of optical spin momentum[51,57] that swirl about the beam axes, inducing corresponding non-conservative forces driving the oscillators out of thermodynamic equilibrium. In addition, light scattering between the particles induces optical binding forces which, in combination with dissipative hydrodynamic interactions, couple their stochastic motion. The relative strength of these coupling interactions varies with the separation, $d$ between the beams, allowing us to tune the form of behaviour manifested in the experiment.

We observe a range of quintessentially non-equilibrium effects ranging from biased stochastic motion to the formation of synchronized limit cycle oscillations, Fig. 1b–d. The experimentally observed stochastic motion can be described in terms of two interconnected pairs of quasi-modes (QMs), whose properties are described in detail below and in the Supplementary Note (III). These pairs of QMs are referred to as the *Centre of Mass* (CoM) and *breathing* (BR) QMs. In combination, they describe in-phase (CoM) and anti-phase (BR) stochastic orbital rotation, in clockwise and counter-clockwise directions, Fig. 1b. Each pair of QMs has a threshold optical power, $P_c$ and $P_b$ for CoM and BR, respectively. Thermal fluctuations combine with non-conservative forces and excite the QMs to different degrees: the closer the optical power is to the threshold power of a QM, the more strongly it is excited and the greater is its mean squared amplitude. In our experiments, we continuously increase the optical power and make observations of the stochastic motion produced. The observed behaviour depends, therefore, on the relative magnitudes of the threshold powers, $P_c$ and $P_b$, see Fig. 1b, c. For example, when $P_c < P_b$, the threshold power of the CoM mode is approached first as the power increases. This causes the CoM mode to grow most rapidly until it dominates the observed motion. When $P_b < P_c$, it is BR that becomes dominant. As described further below, the difference between the threshold powers, $P_c - P_b$, oscillates about zero as the beam separation, $d$, is increased so that $P_c < P_b$ for some separations and $P_b < P_c$ for others. We are therefore able to tune the observed behaviour by adjusting $d$. This dependence on beam separation is shown, for experimental data, in Fig. 1c. Increasing the power above one of the threshold powers results in a sudden bifurcation. Subsequently, each particle executes a limit cycle oscillation. Weak interaction forces cause these self-sustained oscillations to synchronize, see right hand column in Fig. 1d. Sample simulation results, which emphasise the CoM mode are shown in Fig. 1d.

We note that the threshold power is very sensitive to any imperfections in the system. Our numerical stochastic simulation, Fig. 1d assume a perfect system without beam misalignment, aberrations or asymmetries. Under these circumstances, the value of threshold power is (for the experimental value of pressure) an order of magnitude smaller than that observed in our experiments. This has a significant consequence for particle-particle interaction and results in relatively stronger hydrodynamic interactions which favour formation of the CoM QM. Quantitative comparison between theory and experiments was not feasible, see Fig. 1d. A more detailed discussion is provided in section "System sensitivity and connection with experiment".

In the following sections we first outline some theoretical principles before applying them to experimental investigations of the sub-threshold and above threshold regimes (Fig. 1c, d).

### Theoretical considerations: generalized Hooke's law, linear stability and limit cycle formation in stochastic optomechanics

In Supplementary Note (III) we provide a detailed analysis of the general stability properties and stochastic motion of multi-particle,

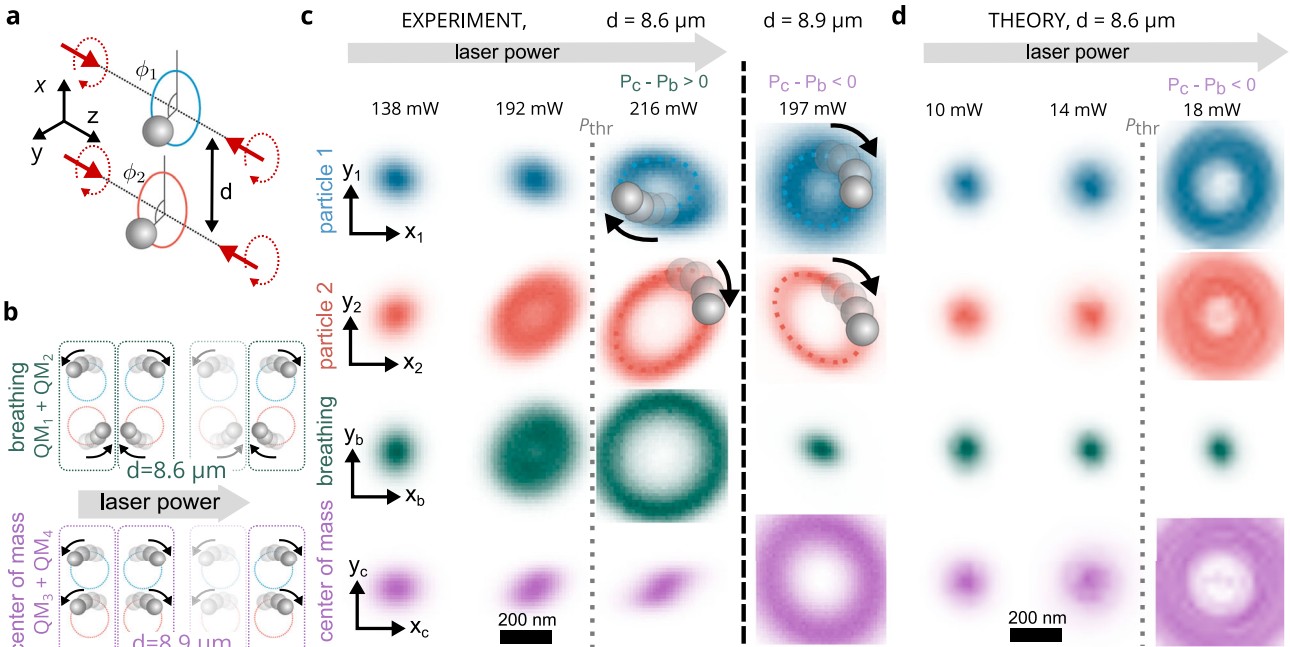

**Fig. 1 | Overview of the experiment. a** Basic geometry and coordinate system. Two pairs of counter-propagating circularly polarized laser beams (red arrows) form two Gaussian standing waves. Two polystyrene particles are localized in the standing waves in the axial direction ($z$-axis), while in the lateral direction, they tend to orbit due to the azimuthal spin force at the ambient pressure. **b** Schematic describing the stochastic quasi-modes in the linear (sub-threshold) regime. Increasing the laser power leads to slightly larger radii of the lateral particles' trajectories and suppression of two quasi-modes QM$_1$, QM$_3$. **c** Experimentally observed biased stochastic motion, showing a tendency towards orbital rotation in the $xy$-plane illustrated using spatial probability densities (PDFs) in the particle displacement basis and in the quasi-mode basis. The picture depicts excitation of the breathing mode QM$_2$ for beam separation $d = 8.6$ μm, corresponding to a

condition where the threshold power $P_c$ for the center of mass mode QM$_4$ is larger than $P_b$ for the breathing mode. For above threshold power (216 mW), the formation of fluctuating limit cycles is illustrated by PDFs with annular distributions. Left-hand/right-hand columns in quasi-mode basis ($x_b y_b$ and $x_c y_c$) show excited breathing ($d = 8.6$ μm)/CoM mode ($d = 8.9$ μm), respectively. **d** Particle motion calculated using stochastic simulation showing similar tendency towards orbital rotation, however, here the threshold power is (for the same pressure) an order of magnitude smaller than for experimental observation and thus the hydrodynamic interaction is relatively stronger than optical, favouring the CoM mode. The threshold power is affected mainly by imperfections and asymmetries in the experimental system. Note, elliptical shapes of PDF and different width of PDF for both particles.

levitated optomechanical systems in the linear regime. Below we summarize the results used throughout the rest of this article.

For many optomechanical systems, including the one studied here, it is possible to identify a configuration in which the system is at mechanical equilibrium, i.e., a configuration in which the external optical forces vanish and are locally restoring. For small displacements, the optical force can be linearly approximated by a generalized Hooke's law, i.e.

$$\mathbf{f} \approx -\mathbf{K} \cdot \mathbf{q} \equiv -P\mathbf{k} \cdot \mathbf{q}, \tag{1a}$$

$$K_{ij} = -\frac{\partial f_i}{\partial x_j}\bigg|_{\mathbf{q}=0}, \tag{1b}$$

where $\mathbf{q} = \mathbf{r} - \mathbf{r}_0$ are small displacements with respect to the coordinates of the mechanical equilibrium, $\mathbf{r}_0$. The stiffness matrix, $\mathbf{K}$, is proportional to the optical power such that $\mathbf{K} = P\mathbf{k}$, with $\mathbf{k}$ the power normalized stiffness.

Two qualitatively distinct cases emerge:

**a. Linearly conservative forces.** In this case $\mathbf{K}$ is symmetric with real eigenvalues. The motion of the system can be described in terms of a discrete, orthogonal set of normal modes, each satisfying the equipartition theorem, having energy $k_b T/2$ for any value of the optical power, $P$.

**b. Linearly non-conservative forces.** In this case, $\mathbf{K}$ is non-symmetric and its eigenvalues can occur in complex conjugate pairs[58]. Pairs of such eigenvalues are associated with *quasi-modes* (QMs) which are not orthogonal, and do not satisfy equipartition. Each QM has characteristic frequencies that can be approximated as,

$$\omega_{i\pm} \approx \pm \sqrt{\frac{P}{m}} \Re\left(\lambda_i^{1/2}\right) + i\left(\pm \sqrt{\frac{P}{m}} \Im\left(\lambda_i^{1/2}\right) + \frac{\xi_i}{2m}\right), \tag{2}$$

where $i$ indexes the QM, $\lambda_i$ is the associated complex eigenvalue of $\mathbf{k}$ and $\xi_i$ is the effective drag, directly proportional to the effective viscosity, $\mu$. In the absence of thermal fluctuations, these frequencies, $\omega_{i\pm}$, relate to damped oscillations in which the coupled coordinates spiral into the fixed point (Supplementary Note (III D)). The rate at which they do so, depends on the imaginary part of $\omega_{i\pm}$, which describes motional damping. By increasing the power, $P$, $\Im(\omega_{i-})$ can be decreased towards zero before changing sign. As it does so, motional damping turns into exponential growth transforming the inward spiral to an outward spiral, destabilizing the trap. The condition for $\Im(\omega_{i-}) = 0$ is,

$$\frac{P}{\xi_i^2} = \frac{\Re(\lambda_i)}{m\Im(\lambda_i)^2}. \tag{3}$$

As the power is increased towards this threshold, the interaction between thermal fluctuations and the non-conservative force causes the instantaneous variance, $\langle a_i^2 \rangle$ and decay time of the autocorrelation

of the QM to increase,

$$\langle a_i(t+\tau)a_i(\tau)\rangle \propto \frac{k_b T}{\Im(\omega_{i-})} e^{i\Re(\omega_{i-})\tau} e^{-\Im(\omega_{i-})\tau}, \qquad (4)$$

where $a_i$ is the amplitude of the QM, and $\Im(\omega_{i-}) \to 0$ (Supplementary Note (III D)). Eventually, the amplitude of the dominant motion exceeds the range over which the forces are approximately linear. Given suitable curvature in the force field, stable limit cycles (i.e. isolated, closed paths in phase space describing self sustained oscillations), or orbits, can form[55] and, ultimately, the fixed point (i.e. the mechanical equilibrium) of the system is destabilized. Experimental observations of such a transition are shown graphically in Fig. 1c, d. As shown in ref. 45 for single oscillators, similar transitions can be induced by variations in pressure, rather than the power. We next apply these principles to our pair of spin-driven oscillators, Fig. 1a.

## Sub-threshold behaviour, optical binding between non-conservative oscillators

First we consider the sub-threshold behaviour, for which the motion remains within the linear range of the force field, where the generalized Hooke's law, Eq. (1) applies, see Fig. 1c. A detailed account is provided in Supplementary Note (IV). The main theoretical results are summarized below and compared with experimental demonstrations. We confine attention to the $xy-$plane, the $z$ motion corresponding to an uncoupled normal mode, satisfying equipartition so that $\langle z^2\rangle = k_B T/Pk_z$, with $k_z$ the stiffness in the $z$ direction. A Gaussian CPCP beam consists of a stack of high intensity planes, each having a transverse Gaussian profile oriented normally to the beam axes. These planes are separated by a spacing of $\Delta z = \lambda/2$. Particles are confined either within these planes or between them with a stiffness varying with size[51]. The particles used in our experiments are strongly localized in the $z$ direction and remain in the same $xy$ plane with variance that decreases with increasing power. In this plane the displacement coordinates are $\mathbf{q} = (\mathbf{q}_1, \mathbf{q}_2) = (x_1, y_1, x_2, y_2)$ and the stiffness matrix for this system have the form,

$$\mathbf{K} = \begin{bmatrix} \mathbf{K}'^{(1)} & \mathbf{A} \\ \mathbf{A} & \mathbf{K}'^{(1)} \end{bmatrix}, \quad \mathbf{K}^{(1)} = \begin{bmatrix} K_r & K_\phi \\ -K_\phi & K_r \end{bmatrix}. \qquad (5)$$

Here $\mathbf{K}^{(1)}$ is the stiffness of a single oscillator, comprising a single sphere in a counter-propagating, circularly polarized trap. The diagonal elements, $K_r$, quantify the stiffness of the purely attractive gradient forces and the off-diagonal terms, $K_\phi$, are connected with non-conservative, azimuthal forces deriving from inhomogeneous optical spin[51]. $\mathbf{K}$ is the stiffness for the pair of oscillators: the stiffness of each constituent oscillator is slightly modified by the proximity of its neighbour, i.e., $\mathbf{K}'(1) \approx \mathbf{K}^{(1)}$, while $\mathbf{A}$ describes the relatively weak coupling between the two particles. Note that $\mathbf{A}$ is, itself, non-symmetric indicating that the interaction is intrinsically non-conservative. Its elements oscillate with the separation between the beams, as is common with conventional binding interactions. A parametric study of the elements of $\mathbf{K}$, and their dependence on beam separation, $d$, and particle radius, $a$, is provided in Supplementary Note (IV A).

The overall form of $\mathbf{K}$ derives from the inversion symmetry of the system, and allows separation into two independent oscillators by transforming to the centre of mass (CoM) and breathing (BR) coordinates ($\mathbf{q}_c$ and $\mathbf{q}_b$ respectively) with,

$$\mathbf{q}_c = (\mathbf{q}_1 + \mathbf{q}_2)/\sqrt{2}, \qquad (6a)$$

$$\mathbf{q}_b = (\mathbf{q}_1 - \mathbf{q}_2)/\sqrt{2}, \qquad (6b)$$

where $\mathbf{q}_{c/b} = (x_{c/b}, y_{c/b})$. This transformation decouples the system stiffness, $\mathbf{K}$, according to,

$$\mathbf{K} = \begin{bmatrix} \mathbf{K}'^{(1)} & \mathbf{A} \\ \mathbf{A} & \mathbf{K}'^{(1)} \end{bmatrix} \to \begin{bmatrix} (\mathbf{K}'^{(1)} + \mathbf{A}) & 0 \\ 0 & (\mathbf{K}'^{(1)} - \mathbf{A}) \end{bmatrix}. \qquad (7)$$

We refer to these two separate oscillators as CoM and BR oscillators. Each has two quasi-modes (QMs), with complex conjugate eigenvalues, which, together, describe stochastic orbital rotation of the coordinates, $\mathbf{q}_c$ or $\mathbf{q}_b$, about the origin. These stochastic motions correspond, respectively, to in-phase (CoM) and anti-phase (BR) circulation of the individual particles, in clockwise or counter-clockwise directions, see Fig. 1b.

Treating the optical and the hydrodynamic interactions as perturbations, Eq. (3) gives the difference between the threshold powers for these oscillators as,

$$P_c - P_b \approx -\frac{\xi_0^2}{m}\left(\frac{4K_r\Im(\delta)}{K_\phi^3} + \frac{9}{2}\frac{a}{d}\frac{K_r}{K_\phi^2}\right), \qquad (8)$$

where $\xi_0 = 6\pi\mu a$ is the Stokes drag on a single particle and $\delta$ is a complex scalar quantity derived from elements of $\mathbf{A}$, that describes the optical interaction. The first term in Eq. (8) is due to optical coupling and the second is caused by differences in the effective drag for the CoM and BR QMs (see Supplementary Note (IV C)). The optical coupling parameter, $\delta$, oscillates with beam separation, $d$, while the hydrodynamic interaction decays monotonically with $d$, serving to systematically reduce the threshold power of CoM relative to BR. As the beam separation, $d$, is increased the CoM and BR oscillators alternately have the lowest threshold power, satisfying $P_c - P_b > 0$ or $P_c - P_b < 0$ (Supplementary Note (IV C)).

At low power all four of these QMs have approximately equal energy and the preference in the sense of stochastic orbital rotation is negligible. As the power is increased, the stochastic motion is increasingly biased towards circulation in the sense dictated by the azimuthal spin forces, although the energy in the CoM and BR QMs remains comparable. For further increases in power, the energy in the QMs with the lowest threshold power begins to grow until it becomes dominant. At this point, the observed motion consists of stochastic rotations of the microspheres around their respective beam axes which are either in phase (CoM dominant), or anti-phase (BR dominant), but always in the direction dictated by the azimuthal spin force.

We investigate these phenomena experimentally in Fig. 1c. Figure 1c shows two dimensional spatial probability distribution functions (PDFs) for the particles. On the top two rows, the PDFs are given in displacement coordinates [i.e. $(x_{1/2}, y_{1/2})$] and, on the lower rows, in the QM coordinates [$(x_{c/b}, y_{c/b})$, Eq. (6)]. These results correspond to a separation of $d = 8.6\,\mu\text{m}$, for which $P_b < P_c$, so that BR grows to dominate, as is clear from the PDFs in the QM basis.

Figure 2 describes the stochastic motion in more detail. Figure 2a–c shows time dependent correlation functions of $x_b$ and $x_c$, for increasing optical power. Written in terms of the components of $\mathbf{q}_{c/b}$ the auto-correlation of these QMs, Eq. (4), is,

$$\langle x_{c/b}(t+\tau)x_{c/b}(\tau)\rangle \propto \frac{k_b T}{\Im(\omega_{i-})}\cos(\Re(\omega_{i-})\tau)e^{-\Im(\omega_{i-}\tau)} \qquad (9a)$$

$$\langle x_{c/b}(t+\tau)y_{c/b}(\tau)\rangle \propto \frac{k_b T}{\Im(\omega_{i-})}\sin(\Re(\omega_{i-})\tau)e^{-\Im(\omega_{i-}\tau)}. \qquad (9b)$$

Equation (9a) describes the increased amplitude and coherence of the stochastically driven oscillations of $x_c$ and $x_b$, while the cross correlation, Eq. (9b), describes the growing tendency of $\mathbf{q}_c$ or $\mathbf{q}_b$ to circulate about the origin[59].

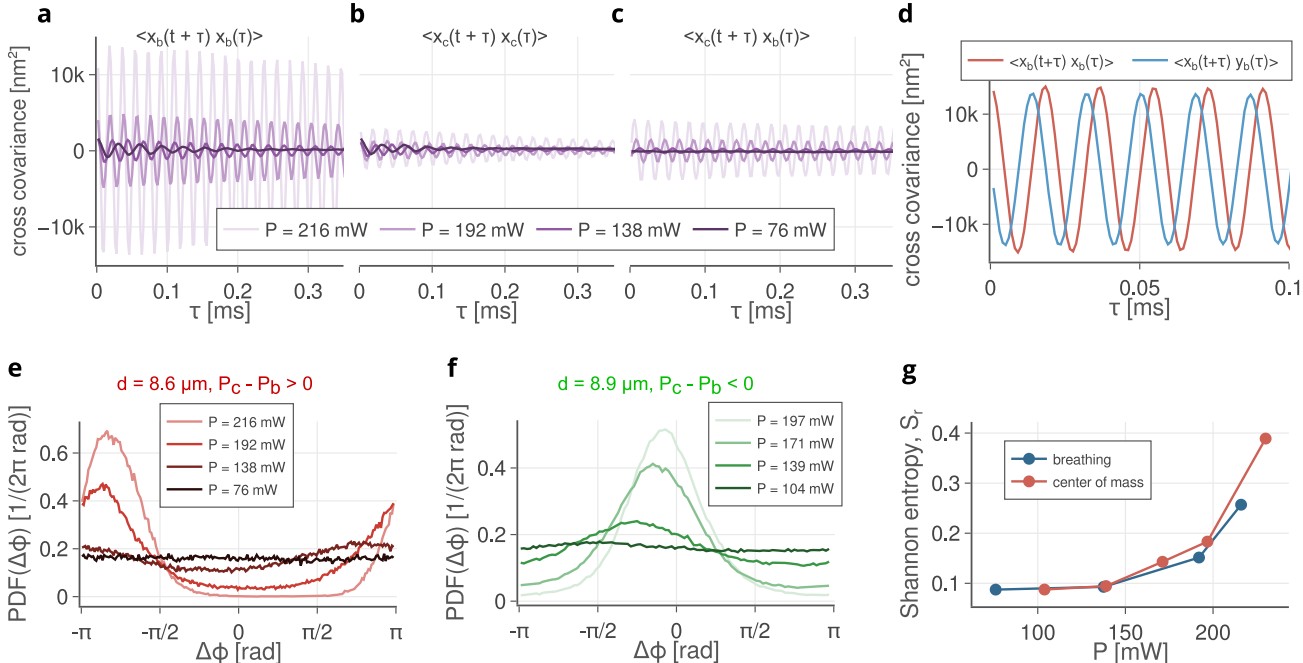

**Fig. 2 | Quantitative analyses of the experimental results below threshold.**
**a** The growth in the auto-correlation of the $x_b$ component of the excited breathing mode $\langle x_b(t+\tau)x_b(\tau)\rangle$ with increasing optical power for the beam distance $d = 8.6\ \mu m$ giving $P_c - P_b > 0$. Increasing the power increases both the mean frequency of the oscillation and the time constant governing the loss of coherence of the oscillation. **b** Due to minor imperfections, the $x_c$ component CoM auto-correlation, $\langle x_c(t+\tau) x_c(\tau)\rangle$, also grows slightly, but the effect is far weaker. **c** The cross-correlation of the $x_{c,b}$ components of CoM and BR, $\langle x_c(t+\tau)x_b(\tau)\rangle$ shows weak coupling caused by minor imperfections in the system. **d** Cross-correlations of the BR coordinates are $\pi/2$ phase shifted, indicating a tendency toward circular motion. **e, f** PDFs of the difference between the azimuthal coordinates of both particles, see Fig. 1a, $\Delta\phi = \phi_1 - \phi_2$ for beam separations of $d = 8.6\ \mu m$ (e) and $d = 8.9\ \mu m$ **f** showing the emergence of phase locking approximating to BR and CoM modes, respectively, shown in Fig. 1e. **g** Relative Shannon entropy $S_r$ as a function of power for the PDFs (**e**) and (**f**).

In Fig. 2a the increase in amplitude and coherence of the BR QM is shown and, in Fig. 2b, the relative stagnation of CoM, see Eq. (9a). The coupling between the CoM and BR oscillators is relatively weak, Fig. 2c, but indicates a slight departure from the ideal symmetry, assumed in Eq. (7), for which CoM and BR motions would be completely independent. The time dependent autocorrelation of $x_b$, $\langle x_b(t+\tau)x_b(\tau)\rangle$, and the cross correlation of $x_b$ with $y_b$, $\langle x_b(t+\tau)y_b(\tau)\rangle$ are shown in Fig. 2d, demonstrating the tendency for $\mathbf{q}_b$ to rotate about the origin, Eqns ((9a),(9b)). This motion corresponds to stochastic rotation of the individual particles about their beam axes, with a relative phase shift of $\pi$ rads[51,59], as illustrated in Fig. 1b.

Figure 2e, f gives an analysis of the statistical behaviour of the azimuthal coordinates of the particles, $\phi_{1/2}$, see Fig. 1a, in the form of PDFs at discrete bins of $\Delta\phi_k = \phi_1 - \phi_2$, $p(\Delta\phi_k)$, as the optical power is increased. For $d = 8.6\ \mu m$, BR grows to dominate the motion, while CoM is emphasized for $d = 8.9\ \mu m$. In Fig. 2g, we plot the relative Shannon entropy, $S_r = 1 - S/S_{max}$, where $S_{max} = \ln N, N$ is the number of bins in PDF, and $S = -\sum_{k=1}^{N} p(\Delta\phi_k) \ln p(\Delta\phi_k)$[60]. In this context, $S_r$ measures synchronization strength, taking values between zero and one, where a value of one indicates perfect synchronization. For the sub-threshold regime, these values of $S_r$ suggest a form of stochastic synchronization, arising prior to limit cycle formation, as the instability is approached.

## System sensitivity and connection with experiment

The treatment given above, for the sub-threshold behaviour of our spin-driven oscillators, provides sound qualitative insight into the behaviour observed in the experiment. However, the physical system is intrinsically sensitive to small departures from ideality and this makes a direct, quantitative comparison difficult to make in this case. The causes of this sensitivity are described in detail in Supplementary Note (VI).

## Above threshold behaviour, synchronization and phase locking of limit cycle oscillators

As a dominant QM grows in amplitude, the particles begin to stray further from the beam axis, where the forces are non-linear, allowing for the formation of self-sustained, periodic trajectories or limit cycles[61]. Eventually the fixed point destabilizes and each particle forms its own limit cycle, resembling a circular orbit. These limit cycles exist independently of one another, and execute a complete cycle in a well defined time period, $T$, with fundamental frequency, $\Omega = 2\pi/T$. The position of the particle on the limit cycle can be associated with a single scalar coordinate, the phase, $\phi$. In our system, two limit cycles are formed, consisting of approximately circular orbits, Fig. 3a.

In general, collections of weakly interacting limit cycles have a tendency to synchronize. That is, their slightly differing fundamental frequencies are drawn together so that the ensemble oscillates collectively with a single, unique frequency[61]. This process is the consequence of small phase adjustments which accumulate over the course of many time periods. In this respect, the mechanisms underpinning synchronization differ fundamentally from those that generate other forms of highly correlated motion as found, for instance, in conservative optomechanics[62], in which the interaction is more direct and the correlation is directly proportional to a coupling constant.

In the mesoscopic regime, synchronization is always accompanied by significant levels of thermal noise. Since limit cycles are neutrally stable (that is, in a single oscillator, each particle is equally stable at any point on the limit cycle), the phase diffuses as it advances. In particular, the total change in phase over a time interval, $\Phi$, has a variance that increases linearly with time[61]. Synchronization of stochastic systems therefore requires that the interaction forces are strong enough to overcome phase diffusion. Nevertheless, fluctuations will still give rise to phase slips, in which the relative phase of oscillators changes abruptly between phase locked states. This

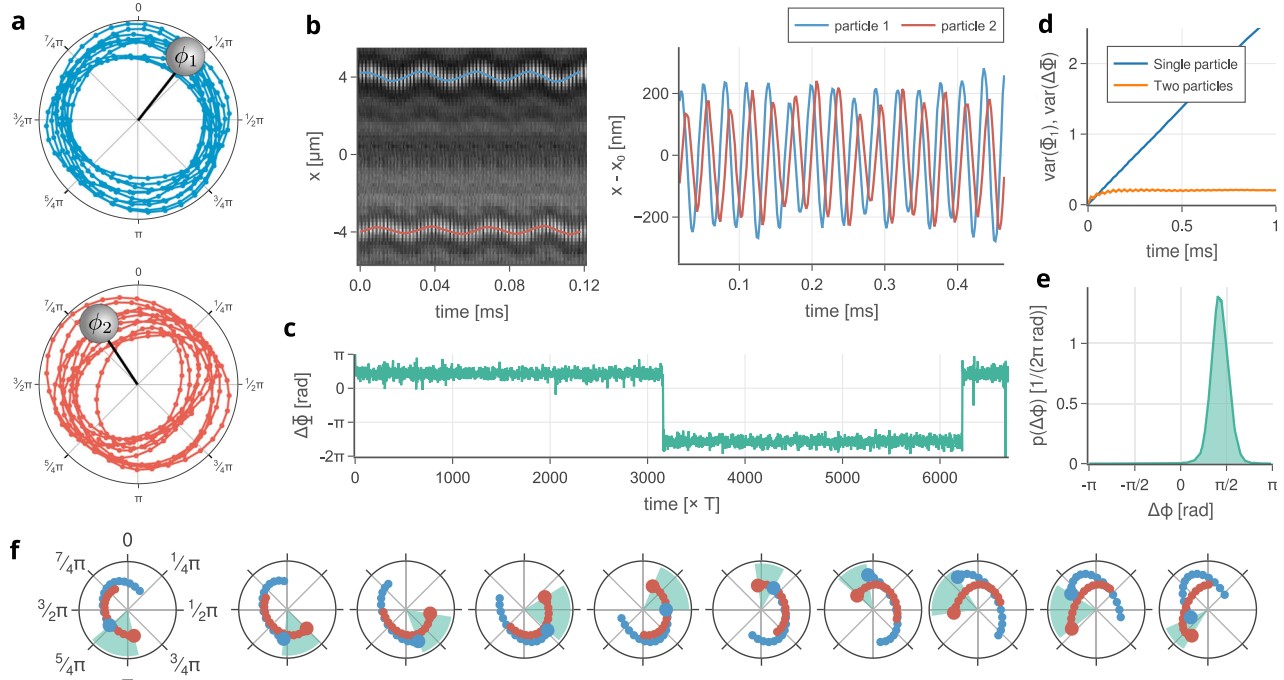

**Fig. 3 | Experimental behaviour of non-conservative oscillators, with the same fundamental frequencies, above threshold where the limit cycles are formed and synchronized. a** Trajectories of both particles in $x - y$ plane corresponding to 10 limit cycle periods. Positions of each particle were detected by an independent quadrant photodiode detector (QPD). **b** The correlated motion of both particles along $x$ axis is visible to the naked eye in records from a fast CMOS camera plotted against time. Blue and red curves link the centers of particle images. **c** Accumulated phase difference, $\Delta\Phi = \Phi_1 - \Phi_2$, plotted against time, shows discrete phase jumps between phase-locked states. **d** Experimental demonstration of the phase diffusion properties comparing the linear time dependence of the variance of the accumulated phase of a single oscillator i.e. Var($\Phi_1$) (blue) with the variance in the accumulated phase difference between the synchronized oscillators, i.e., Var($\Delta\Phi$). **e** Probability density function PDF of the relative phase $\Delta\phi$ of the oscillators obtained by a fast CMOS camera and mapped on the interval $\langle -\pi, \pi \rangle$. **f** Short time trajectories of particles demonstrating in-phase synchronized motions.

happens when fluctuations push the oscillators far out of the synchronized state and, instead of reversing back into it, synchrony is restored after one of the oscillators when the phase of one oscillator increases by $2\pi$ relative to the other[61]. This can be likened to Kramers hopping in a potential[63,64] although, in this case, the transitions take place between non-equilibrium states and a closer analogy is with stochastic motion in a tilted periodic potential[65,66].

The above threshold synchronization of our twin oscillators is explored experimentally in Figs. 3 and 4. Accompanying simulations and theoretical comments are provided in Supplementary Note (V). We take the azimuthal coordinate of the particle displacement as the phase of the oscillator. This is less rigorous than the definition obtained from phase reduction but is far simpler to describe and sufficiently accurate to capture the required phenomena (see Supplementary Note (V B)). In the following discussion we distinguish between the absolute phase of an oscillator, $\phi_i$ with $i = 1, 2$, which specifies the position of particle $i$ on its limit cycle and takes values in the interval $(-\pi, \pi)$, and the accumulated phase, $\Phi_i$, which specifies the total phase difference covered in a given period of time. Associated with these quantities are the absolute phase difference between the oscillators, $\Delta\phi = (\phi_1 - \phi_2)$ [restricted again to the interval $(-\pi, \pi)$] and the accumulated phase difference, $\Delta\Phi = (\Phi_1 - \Phi_2)$. Figure 3 describes an established synchronized state for oscillators with approximately equal fundamental frequencies. Experimentally measured trajectories are depicted in Fig. 3a, b, f. The accumulated phase difference shows, in Fig. 3c, a typical noise-induced phase slip. Figure 3d confirms the established linear relationship between the time dependence of the variance of the accumulated phase, $\Phi_1$, for a single oscillator (blue line). In contrast, the variance in the difference of the accumulated phases of a pair of synchronized oscillators saturates quickly

demonstrating that the interactions are strong enough to suppress phase diffusion in the synchronizing pair. The steady-state PDF of $\Delta\phi$ appears in Fig. 3e, showing a sharply peaked phase difference with relative Shannon entropy $S_r = 0.39$.

The detuning behaviour obtained when the limit cycles of the oscillators have differing fundamental frequencies is described in Fig. 4. Figure 4a–c shows the effect of varying the second beam waist radius between $w_0 = 1.03\,\mu m$ and $w_0 = 1.06\,\mu m$, holding the first constant at $w_0 = 1.03\,\mu m$. This has the effect of continuously varying the fundamental frequency of the second limit cycle oscillator (see ref. 51). Figure 4a, b shows the accumulated phase difference, $\Delta\Phi$, for a series of detuned oscillators over different time intervals. Over long times, Fig. 4a shows a steady increase in the accumulated relative phase of the oscillators as the faster oscillator pulls ahead of the slower. Figure 4b shows the detailed motion over shorter time scales. As described previously, the oscillators synchronize perfectly for short intervals before fluctuations induce a phase slip of $2\pi$ radians, or an integer multiple. As the detuning increases the time between phase slips decreases until synchronization becomes impossible. The effect on $p(\Delta\phi)$ is to lower and broaden the main peak, shifting it to slightly greater phase differences on average, Fig. 4c. We note that this broadening is in favour of the faster oscillator, and may reflect the growing difference in the driving forces connected with detuning. Over the course of this variation the Shannon entropy changes from 0.35 to 0.17, Fig. 4d.

These experimental results are supported by dynamical simulations (see Supplementary Note (V)), which shed light on the synchronization mechanism. In particular, optical interactions alone cannot account for the observed effects. Dissipative, hydrodynamic interactions act cooperatively to generate and stabilize synchronized states.

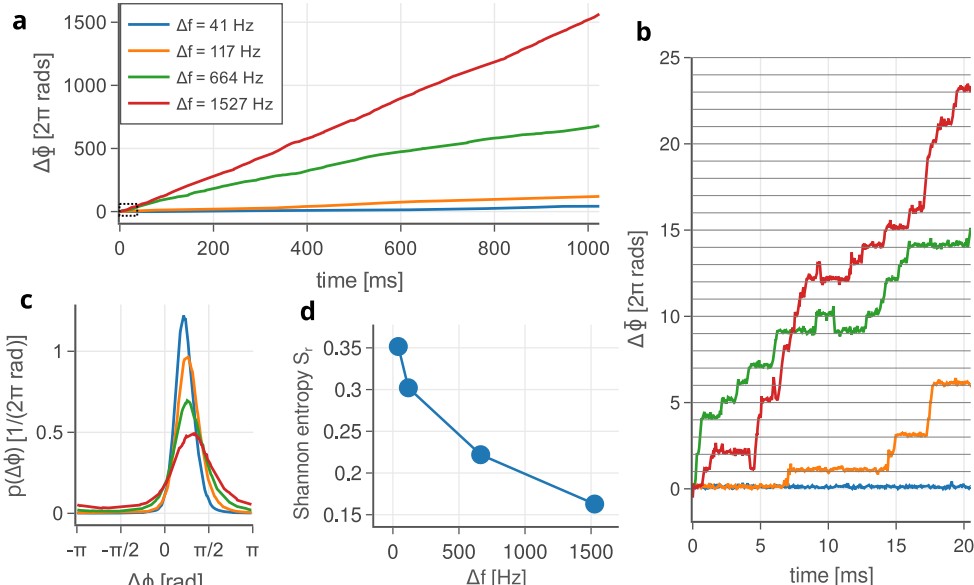

**Fig. 4 | Behaviour of the relative phase of two interacting limit cycle oscillators having detuned fundamental frequencies** $\Delta f = f_2 - f_1$. **a** Long-time traces of the accumulated phase difference $\Delta\Phi$ for a series of detuned oscillators. **b** Shorter-time traces of records **a** illustrating phase slips of an integer multiple of $2\pi$ radians. As the detuning increases the time between phase slips decreases until synchronization becomes impossible. **c** Broadening of the PDF of the absolute phase difference, $\Delta\phi$ for larger detuning. **d** Quantitative characterization of weakening of synchronization using the relative Shannon entropy $S_r$.

## Discussion

In this article we have described the emergence of archetypal, non-equilibrium behaviour in a non-conservative optomechanical system consisting of a pair of non-Hermitian oscillators driven by optical spin momentum.

We have shown that stochastic motion becomes progressively more biased, coherent and deterministic as the optical power is increased. Particular forms of motion, described by quasi-modes, begin to dominate. Further increases in power result in collective a Hopf bifurcation and the formation of limit cycle oscillations which interact and synchronize. This general behaviour is representative of a far wider class of systems than the particular example dealt with here.

In addition, our results suggest that hydrodynamic interactions could play a role in the formation of coordinated motion in both the linear and non-linear regimes. The dependence of hydrodynamic coupling, and therefore dissipation rate, on the configuration of the system appears to influence the formation of these non-equilibrium steady states. This effect could be analogous to the minimal dissipation principle of Onsager[67]. We note that hydrodynamic coupling in this regime is a relatively unexplored issue. However, its influence has been inferred experimentally in similar systems[53,62]. In the case of these works, the Oseen tensor captures the qualitative effect but appears to underestimate its magnitude. These considerations imply a fundamental difference between this system, and the paradigmatic, Kuramoto model for synchronization[28], in which the underlying mechanism relies on reactive forces alone.

More generally, the combination of structured non-conservative forces and coupled dissipation open up numerous new themes for continuing research in levitational optomechanics. These avenues range from the development of novel forms of mesoscale topological matter[36,38], to the experimental exploration of emerging and controversial issues in the stochastic thermodynamics, of non-equilibrium states, such as the synchronized states described here. Application of the cooling protocols, previously applied to conservative systems, could even push these effects towards the quantum regime allowing experiments to probe the quantum-classical interface for dynamic phenomena such as limit cycle formation or synchronization.

A proposed route to the quantum regime is described in Supplementary Note (VIII).

## Methods
### Experimental details

In order to optically confine the particles inside a vacuum chamber and characterize their optical binding, we used a source of infrared laser light operating at the vacuum wavelength of 1064 nm with low intensity noise (Coherent Mephisto). We used Thorlabs achromatic doublets with antireflection coating ACN254-XXX-C (L1 – L6), dielectric mirrors PF10-03 (M1 – M3) and aspheric lenses C240TME-C with antireflection coating (AS1).

A collimated Gaussian beam from an infrared laser was expanded by a telescope formed by lenses L1 ($f_1 = 150$ mm) and L2 ($f_2 = 300$ mm) and projected on a spatial light modulator (SLM) (Hamamatsu LCOS X10468-07). The phase mask encoded at the SLM diffracted the beam into the ±1 diffraction orders that were used to generate the two counter-propagating trapping beams; the zeroth and higher orders were blocked by a stop placed in the focal plane of lens L3 ($f_3 = 400$ mm).

The two transmitted 1st-order beams were reflected from prisms P1 and collimated by lenses L4 ($f_4 = 200$ mm). These lenses formed telescopes with the lens L3, projecting the SLM plane on the mirrors M2. The SLM plane was then imaged onto the back focal planes of aspheric lenses AS1 ($f = 8$ mm, maximal NA = 0.5) by telescopes consisting of lenses L5 ($f_5 = 100$ mm) and L6 ($f_6 = 150$ mm).

Two pairs of horizontal counter-propagating laser beams generated by splitting a single incident beam with a spatial light modulator (SLM) were focused inside the vacuum chamber by two aspheric lenses with NA = 0.5, leading to the beam waist radii $w_0$ adjustable in the range 1–3 μm. The focal planes of the four beams created in the trapping region were slightly displaced from each other along the beam propagation direction $z$ (by ~5 μm, see red lines in Fig. 5) to increase the axial trapping stability[68] (Fig. 5).

Widths of the focused trapping beams in the sample chamber could be controlled by adjusting the area of the diffraction grating imposed upon the SLM.

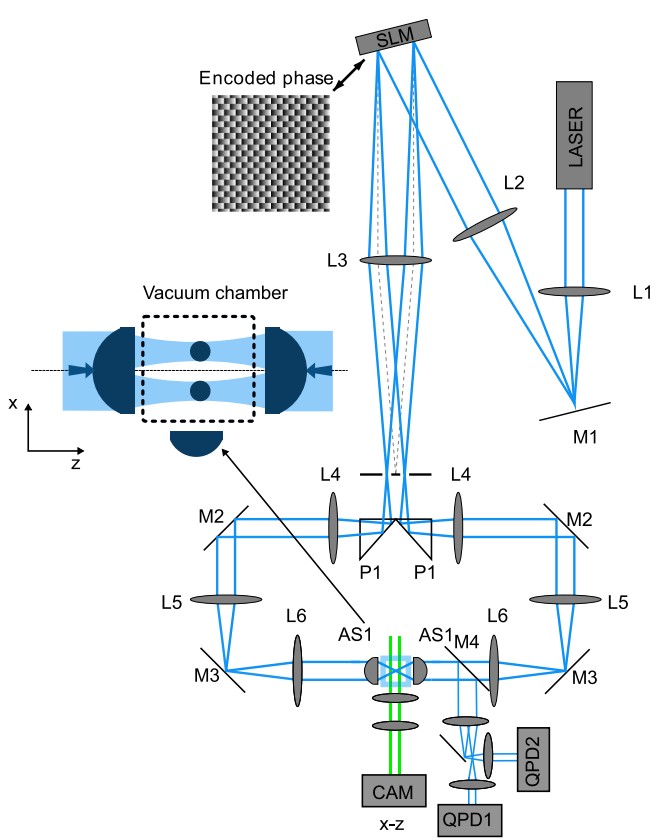

**Fig. 5 | Experimental set-up of two pairs of counter-propagating beams forming standing wave optical traps with circularly polarized light.** Particles are trapped in a small vacuum chamber, dashed square in the inset, placed between the focusing aspherical lenses AS1,2. Positions of particles in $x-z$ plane are magnified by a telescope and observed by CAM1. Positions of each particle in $x-y$ plane are independently but synchronously recorded by quadrant photo-detectors QPD1,2.

Polystyrene particles (Polysciences, mean diameter 850 nm) were dispersed in isopropyl alcohol and after ~20 min sonication of the suspension, droplets containing the particles were sprayed into the trapping region in the vacuum chamber employing an ultrasonic nebulizer (Beurer IH 50).

We employed two quadrant photo diodes to record the motion of the particles. Trajectories were recorded for durations of 2 s with 1 MHz sampling frequency. At the same time we used a fast CMOST camera (I-speed 5 series from IX Camera, exposure time was set to 1 $\mu$s and the frame rate was 300 kHz) to record the motion of the particles in $x-z$ plane.

To enable position tracking of the optically trapped and bound particles, the sample was illuminated by an independent laser beam (Coherent Prometheus, vacuum wavelength 532 nm) propagating along the $y$-direction perpendicular to the imaging $xz$-plane. Large beam waist radius $w_0 = 40\,\mu$m and low power (~5 mW at the sample) of the green illuminating beam ensured its negligible contribution to the net optical force acting on the particles. Typically, we recorded at least 100,000 frames from the studied optically bound structures to obtain sufficiently long trajectories for the analysis of their motional dynamics.

The off-line tracking of the particle position from the high-speed video recordings was based on the determination of symmetries in the particle images[69]. Briefly, since a spherical particle produces an azimuthally invariant image, we used the shift property of the Fourier transform and looked for the best horizontal and vertical symmetries

in the particle image, which provided us with the information about the in-plane $x$ and $z$ coordinates.

## Data availability
All data that support the findings of this paper have been deposited to the Zenodo repository https://doi.org/10.5281/zenodo.8220240.

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

## Acknowledgements

The Czech Science Foundation (GF21-19245K, O.B.); Akademie věd České republiky (Praemium Academiae, P.Z.); Ministerstvo Školství mládeže a tělovýchovy (CZ.02.1.01/0.0/0.0/16_026/0008460).

## Author contributions

S.H.S., O.B., and P.Z. designed and developed the study from the the-oretical and experimental aspects, S.H.S. provided theoretical content, O.B., M.D., P.J., and J.J upgraded the experimental setup and performed

the measurements, S.H.S., O.B., M.Š. and P.Z. analysed the experimental data and compared them to the theoretical results. S.H.S., O.B., and P.Z. contributed to the text of the manuscript.

## Competing interests

The authors declare no competing interests.
