## [Peer Review file · Nature Communications]

REVIEWER COMMENTS

Reviewer #1 (Remarks to the Author):

This manuscript describes theoretical and experimental studies of cooperative motion of colloidal spheres trapped by circularly polarized optical tweezers in a low-pressure gas. The particles experience parallel torques arising from the light's angular momentum that cause them to orbit their traps' centers. Under weak-coupling conditions, the particles undergo Brownian motion in their traps, and their fluctuations are biased into circulation by the optical torque. Increasing laser power also increases coupling due to optical and hydrodynamic interactions. The particles undergo a transition to deterministic circular trajectories and their orbits become synchronized. Depending on the strength and nature of the coupling, the synchronized state favors one of two quasi-normal modes: a breathing mode in which the center of mass is stationary or a center-of-mass mode in which the center of mass traces out a circular orbit. Synchronization is strongest when the two particles' orbits have the same natural frequency. Detuning the natural frequencies leads to phase-slipping and a loss of coherence.

This is an interesting system and is a suitable subject for Nature Communications. The present study carves out new territory by working in an inviscid medium so that inertia contributes to the systems' dynamics. Previous studies on related systems involve particles that are dispersed in water whose motions are strongly overdamped. Having said that, the manuscript largely overlooks previous work on coupled colloids in arrays of optical traps and would be improved by placing itself more clearly in that context.

The coupled dynamics of colloidal spheres in optical traps originally originally was discussed in

* Meiners JC, Quake SR. Direct measurement of hydrodynamic cross correlations between two particles in an external potential. *Physical Review Letters* 82, 2211 (1999)

* Polin M, Grier DG, Quake SR. Anomalous vibrational dispersion in holographically trapped colloidal arrays. *Physical Review Letters* 96, 088101 (2006)

The latter of these studies clarified how hydrodynamic coupling selects normal modes in thermally driven systems.

These pioneering studies inspired subsequent research into synchronization in hydrodynamically coupled colloidal arrays under external driving, with examples including

* Kotar J, Leoni M, Bassetti B, Lagomarsino MC, Cicuta P. Hydrodynamic synchronization of colloidal oscillators. *Proceedings of the National Academy of Sciences* 27, 7669 (2010)

* Koumakis N, Di Leonardo R. Stochastic hydrodynamic synchronization in rotating energy landscapes. *Physical Review Letters* 110, 174103 (2013)

* Kotar J, Debono L, Bruot N, Box S, Phillips D, Simpson S, Hanna S, Cicuta P. Optimal hydrodynamic synchronization of colloidal rotors. *Physical Review Letters* 111, 228103 (2013)

* Juniper MP, Straube AV, Besseling R, Aarts DG, Dullens RP. Microscopic dynamics of synchronization in driven colloids. *Nature Communications* 6, 7187 (2015)

* Maestro A, Bruot N, Kotar J, Uchida N, Golestanian R, Cicuta P. Control of synchronization in models of hydrodynamically coupled motile cilia. *Communications Physics* 1, 28 (2018)

Several of these publications discuss the theory of hydrodynamic synchronization. It would be helpful for the present contribution to clarify its novel contributions relative to this earlier work. One principal distinction is that the present study is performed at high Reynolds number whereas the previous studies all were carried out at low Reynolds numbers.

I believe that this text is the first to use the term “quasi-mode” to describe non-equilibrium states arising from conjugate pairs of eigenvalues of the stiffness matrix. If this term has been used previously, a reference should be provided. The power-dependent transition that quasi-modes make to underdamped oscillation is reminiscent of the scale-dependent crossover to underdamped propagation reported by Quake and collaborators in the thermally-driven array.

I found Fig. 1 difficult to understand until I had read the entire paper a couple of times. I still am not entirely sure what the blurs in panels (c) and (d) represent. I recommend using different colors for the PDFs of the two particles.

Figure 2 is difficult to read. The text is microscopic and the superimposed curves in (a), (b) and (c) cannot be distinguished.

Reviewer #2 (Remarks to the Author):

Synchronization is a hallmark non-equilibrium effect that is ubiquitous in nature, with very clear implications for probing topological effects and sensing. Yet it has remained elusive to investigate this phenomenon in the mesoscopic domain.

The authors present here a careful and detailed study of precisely such a system using optically levitated microspheres in vacuum. They explore a wide parameter range that investigates the out-of-equilibrium dynamics of their coupled oscillators, leading to smoking-gun signatures of synchronization in Fig 3. I particularly appreciate the care taken to verify their results with different analysis - the difference in variance between the single/two particle case, measuring the Shannon entropy as well as through the probability density function of the accumulated phase between the oscillators. They further investigate the breakdown of synchronization with the frequency detuning between the oscillations.

The methods are technically sound, the results are analysed and interpreted carefully, and presented in great detail. The main claim of synchronization is well supported by the data. The results of this manuscript are highly relevant not only to the field of levitated optomechanics, but also to the wider community probing non-equilibrium phenomena across disciplines. I have no doubt that these scientific results meet the criteria for publication in Nature Communications.

I will be happy to recommend the manuscript for publication if the following comments are addressed satisfactorily:

(1) The authors disregard z motion as an uncoupled normal mode. What is the justification for this, given that the motion is not cooled, leading to high amplitude oscillations corresponding to room temperature?

(2) In Fig 2b/c, the coupling seen in the autocorrelation of the CoM mode and the cross-correlation of the CoM/BR modes is described as being "due to minor imperfections". Could the authors provide more insight into the types of imperfections and how they might lead to such coupling?

(3) The author's claim of "Application of the cooling protocols... could push these effects towards the quantum regime" does not seem to follow from the paper, as any cooling protocol applied to their system would simply suppress orbital motion of the particles. Could they elucidate on what exactly is meant by this statement?

(4) I was disappointed to see that despite the detailed theoretical simulations in the supplementary, there are no comparisons (even qualitative) between the experimental and theoretical results in the figures. This could provide validation and potential insight into the relative contributions of the hydrodynamic and optical interactions to the measured data, which is a central theme of the paper. Why have the authors chosen to not show theoretical plots in the main figures?

(5) The Discussion section emphasises that hydrodynamic interactions play a key role in synchronisation. This may be true, but there is no convincing evidence for this in the experimental data. In fact, every mention of hydrodynamic interactions in the main text is in conjugation with optical interactions, with no attempt to isolate hydrodynamic contributions. The theoretical simulations in the supplementary indicate that such interactions play an important role (and might even be sufficient) for synchronization, but without a connection to the main experimental figures (see comment #4), I do not think these theoretical results are sufficient to warrant the strong claim in the Discussion session about the role of hydrodynamic interactions.

(6) I appreciate the care taken by the authors to calculate the Shannon entropy as an estimate of the synchronization strength. However, this inevitably raises questions in the reader about why, even well above threshold, the Shannon entropy is < 0.4 , which suggests that the synchronization is still stochastic.

(7) Can the authors provide possible mechanisms for the "shift to slightly greater phase differences" in reference Fig 4c?

Minor comments:

8) Fig 1a caption: Typo "fromve".

9) Fig 1d caption: "further developed stable" and "further grows" are hard to understand.

10) Sec 2a text, paragraph 2: Two references to Fig 1f, which does not exist.

11) Sec 2c text, first paragraph on page 6: "in for" should be "for".

12) Sec 2D, brackets not closed in references in supplementary.

13) Sec 3 last paragraph: "experiment" should be "experiments"

14) The description of what a limit cycle is, given in page 4, needs to appear in the introduction.

15) Statements that need to be expanded for readability: (a) "Since limit cycles are neutrally stable,...", unclear what "neutral" means, and (b) "a typical noise-induced phase slip", needs either a reference or a description of the mechanism leading to such phase slips, especially because they become relevant later in Fig 4.

Reviewer #3 (Remarks to the Author):

The manuscript by Brzobohaty et al describes experiments and a corresponding model involving optically trapped particles interacting with their trapping beams and each other, ultimately resulting in synchronized motion. Such motion involves a variety of forces and phenomena, and I applaud the authors for putting in the significant effort required to understand and explain the mechanics behind their observations. However, I do have several comments and concerns about the manuscript in its current form.

First, I have a few suggestions for minor changes in the writing:

1. Several important details are only specified in figure 1, while (in my opinion) they should also be in the main text for clarity. These are the ambient pressure and the parameters that set the size scale (particle size, beam waist, and beam separation).

2. Near the end of page 3, I think the word "the" should be dropped from "... can be decreased towards zero before the changing sign."

Next, I have some scientific concerns:

1. The ambient pressure at which the experiments take place (17 mbar) is quite high for typical levitated optomechanics experiments, particularly those that have any interest in approaching the quantum regime (which typically utilize pressures 7 to 10 orders of magnitude lower). At 17 mbar, it seems that any interesting quantum state would decohere on extremely short timescales (likely much shorter than any other timescale in the reported experiments), but the possibility of using "cooling protocols" to push these experiments into the quantum regime is suggested (although primarily only in the Introduction and Discussion sections). Is there any plausible path to quantum behavior from these experiments? If it would require more than cooling, that should be noted.

2. The ambient pressure is only stated in one place (in the figure 1 caption), but this seems to be a critical parameter of the system based on the analysis in the supplemental information. How sensitive are the experiments to the ambient pressure? Was similar behavior observed over some pressure range (although presumably with different optical powers), or does the pressure need to be carefully tuned?

3. In the analysis of the effects of the ambient pressure in the supplemental information, it appears that the only effect considered is drag (including coupling between the particles due to drag). Were radiometric forces (due to the interaction of thermal gradients and gas) considered? Is there justification for ignoring these forces? Anecdotal evidence suggests that radiometric forces may be very important at these size scales and at high pressure (near atmospheric pressure). A quick estimate suggests a mean free path of gas molecules of ~ 4 micrometers at 17 mbar and room temperature, quite close to the diameter of the particles (~ 1 micrometer) and thus in the regime where radiometric forces could be significant. It is still quite plausible that the speed of the particles is so high that drag forces are dominant, but I did not see any discussion of this in the manuscript.

Overall, the manuscript reflects a significant effort to understand complex behavior in a deceptively simple system. With some potentially minor clarification and revision, I feel the manuscript could be deserving of publication.

Response to reviews

I. REVIEWER #1

A. Major Issue

Reviewer #1 is primarily concerned with the connection between the system described in our article, with the systems previously studied in relation to hydrodynamic synchronization at low Reynolds (low Re) number. In particular, reviewer #1 is concerned that we have overlooked the literature on low Reynolds number hydrodynamic synchronization. The editor interprets the comments of reviewer #1 as a request that our work be *in a proper context with respect to the existing literature on the coupled dynamics of colloidal spheres in optical traps and their 'hydrodynamic synchronization' under external fields.*

In response to reviewer #1, we are only too happy to describe the relationship between our system and low Re systems, and to include appropriate references. Indeed, some of the authors involved with the article under review were co-authors on the articles mentioned by the reviewer! As described below, we have included a paragraph in the main text of the paper, which discusses this issue, and which refers to an additional section added to the Supplementary Information which provides a more detailed account.

However, we note that our system is, in many respects, **more** general than the low Re systems studied previously. Indeed, the interest in low Re systems could be said to derive from the restricted nature of these systems - low Re work seeks to answer the question 'can hydrodynamic coupling alone give rise to synchronization at low Reynolds number'. In contrast, our system involves inertia and the synchronization mechanism features coupling in both reactive (i.e. optical) and dissipative (i.e. hydrodynamic) forces. Another fundamental difference between our work and the previous work at low Re is that, in our system, the paths followed by the particles are spontaneous; they are determined by the underlying stochastic equations of motion. In contrast, much (maybe all?) of the work at low Re makes use of prescribed trajectories and modulates force profiles in a prescribed manner to optimise or tune synchronization. Moreover, synchronization is a generic phenomena, known to occur from astrophysical to microscopic length scales. While the relationship with low Re hydrodynamic synchronization is indeed interesting, one could, with equal or greater validity, draw comparisons between our work and synchronization in planetary systems (for example) which, like our system, are subject to inertia and synchronize through reactive forces (gravity). We have not, therefore, presented our system *in the context of* low Re systems, as suggested by the editor, since our system is (i) more general, and (ii) perhaps no more strongly related to low Re systems than to myriad other synchronizing systems - especially those involving inertia and reactive coupling.

a. *Specific changes to our article:* In the main text, we have included the following paragraph to the Introduction, briefly comparing our system with low Re hydrodynamic synchronization:

Similar systems have been studied extensively in the low Reynolds number (low Re) regime, see [36]. The significance of these low Re systems lies in their application to micro-biology, and on their reliance on coupled dissipative forces (i.e. hydrodynamic coupling) to achieve synchronization. The system we study here is, in many respects, fundamentally different: (i) our system is underdamped, with steady state conditions formed by a delicate balance between (reactive) optical, (dissipative) hydrodynamic and inertial forces, (ii) coupled reactive forces play a key role in synchronization, and (iii) our system is completely unconstrained, spontaneous and autonomous, in comparison with model low Re systems, in which both the paths followed by the particles, and the force profiles that drive them, are prescribed by the experimenter [37,38]. This final point is important. In contrast, the stochastic trajectories followed by the particles in our system derive from underlying physical principles, and so can be used to test emerging concepts in stochastic thermodynamics such as thermodynamic uncertainty relations [39]. These issues are discussed further in Supplementary Note (VII).

Where the new references are,

[36] J. Elgeti, R. G. Winkler, and G. Gompper, Rep. Prog. Phys. 78, 056601 (2015).

[37] J. Kotar, L. Debono, N. Bruot, S. Box, D. Phillips, S. Simpson, S. Hanna, and P. Cicuti,

Phys. Rev. Lett. 111, 228103 (2013).

[36] A. Maestro, N. Bruot, J. Kotar, N. Uchida, R. Golestanian, and P. Cicuti, Commun. Phys. 1, 28 (2018).

[37] S. Lee, C. Hyeon, and J. Jo, Phys. Rev. E 98, 032119 (2018).

Reference [36] is a review paper citing almost all of the work highlighted by the reviewer. Supplementary Note (VII) contains a more detailed discussion in which we individually cite the work highlighted by the reviewer.

B. Minor Issues

Finally, Reviewer #1, comments:

1. I believe that this text is the first to use the term "quasi-mode" to describe non-equilibrium states arising from conjugate pairs of eigenvalues of the stiffness matrix. If this term has been used previously, a reference should be provided.

We agree with the reviewer; to the best of our knowledge this is the first text to use the term "quasi-mode" in this sense. Quasi-normal modes appear in the physics of black holes, where they also relate to conjugate pairs of eigenvalues. However, they do not describe non-equilibrium states as they do here. We selected the term "quasi-mode" since these modes are not true modes (in particular, they are not orthogonal and, when used to determine correlation functions, for example, this property needs to be handled carefully, as we show in the Supplementary Information). In comparison, the term "quasi-normal mode" suggests that we are referring to something that is a true mode, and that it is almost normal - but not quite. We therefore prefer to stick with the term "quasi-mode". No additional references have been added since we believe that this is the first article to coin the term.

2. The power-dependent transition that quasi-modes make to underdamped oscillation is reminiscent of the scale-dependent crossover to underdamped propagation reported by Quake and collaborators in the thermally-driven array.

We thank the reviewer for calling our attention to this analogy. We have added a reference to this article to the Supplementary Information, section II:

Parenthetically, we note the analogous treatment for the low Reynolds number case, where the Oseen tensor captures the required dynamics for hydrodynamically coupled spheres in an external potential [6,7].

Where the new references are,

[6] M. Polin, D. G. Grier, and S. R. Quake, Phys. Rev. Lett. 96, 088101 (2006).

[7] J.-C. Meiners and S. R. Quake, Phys. Rev. Lett. 82, 2211 (1999).

After due consideration we decided not to refer to this paper in connection with the power-dependent transition we observe for quasi-modes. The reason for this is that the cross-over

effect proposed by Quake et al. is fundamentally different, and may confuse the reader. In particular, the motion connected with Quake's cross-over effect is stable and conservative - whereas the effect we document is connected with a non-conservative instability.

3. I found Fig. 1 difficult to understand until I had read the entire paper a couple of times. I still am not entirely sure what the blurs in panels (c) and (d) represent. I recommend using different colors for the PDFs of the two particles.

We have clarified the text in the figure caption that describes panels (c) and (d) - and we've used different colors for the particle PDFs, as suggested.

4. Figure 2 is difficult to read. The text is microscopic and the superimposed curves in (a), (b) and (c) cannot be distinguished.

We have improved the readability of the figure in both respects.

II. REVIEWER #2

We thank the reviewer for their opening remarks, for endorsing the main claim of the paper and for their comments regarding our article's suitability for publication in Nat. Comms. (given appropriate responses and revisions). Our point-by-point response follows.

1. (1) The authors disregard z motion as an uncoupled normal mode. What is the justification for this, given that the motion is not cooled, leading to high amplitude oscillations corresponding to room temperature?

The justification for ignoring motion in the z direction is, in the first instance, empirical; the z widths of the centre of mass of each particle are tightly confined has been addressed in previous papers; counter-propagating circularly polarized beams form stacks of interference fringes oriented normally to the beam axes and separated by half the wavelength. Between these interference fringes, the intensity falls to zero. Superposed onto these interference fringes is a much broader, more slowly varying envelope imposed by the Gaussian beam profile. The maximal intensity gradients are, therefore, in the z direction, associated with the interference fringes. The stiffness for dielectric spheres, held in these beams, depends on the ratio of the sphere size to the fringe spacing. When this ratio is an integer, the stiffness is low, otherwise it is often the highest stiffness in the system. In our case, the z stiffness is high. Since the z motion is uncoupled to first order, the variance of the z motion satisfies equipartition, i.e. $\langle z^2 \rangle = kBT/Pkz$, for power, P , and normalized stiffness, kz . Thus, as we increase the power, the confinement in the z direction increases. Both in the experiment and simulations, the particles are always tightly confined in the z direction. Finally, we note one related issue. As is clear from the preceding discussion, the particles are confined within (or between) interference fringes. Although the Gaussian

envelope generally leads to the particles being held in fringes with the same z coordinate, this does not appear to be critical for synchronization. This, perhaps, should not be surprising since synchronization is a generic phenomena.

To clarify this issue we have added the following text to Section IIC in the main text.

We confine attention to the xy -plane, the z motion corresponding to an uncoupled normal mode, satisfying equipartition so that $\langle z^2 \rangle = k_B T / P k_z$, with k_z the stiffness in the z direction. A Gaussian CPCP beam consists of a stack of high intensity planes, each having a transverse Gaussian profile oriented normally to the beam axes. These planes are separated by a spacing of $\Delta z = \lambda/2$. Particles are confined either within these planes or between them with a stiffness varying with size [31]. The particles used in our experiments are strongly localized in the z direction and remain in the same xy plane with variance that decreases with increasing power.

2. (2) In Fig 2b/c, the coupling seen in the autocorrelation of the CoM mode and the cross-correlation of the CoM/BR modes is described as being "due to minor imperfections". Could the authors provide more insight into the types of imperfections and how they might lead to such coupling?

The short answer is that the beams are slightly aberrated. Although we attempt to correct aberrations with the spatial light modulator, weak reflections from the glass windows of our small vacuum chamber induce additional aberrations we are unable to eliminate. This issue is related to point (4), described below in more detail.

3. (3) The author's claim of "Application of the cooling protocols... could push these effects towards the quantum regime" does not seem to follow from the paper, as any cooling protocol applied to their system would simply suppress orbital motion of the particles. Could they elucidate on what exactly is meant by this statement?

The reviewer is absolutely right to raise this point, although we note that this is a speculative suggestion we make, rather than a scientific claim. Nevertheless, we believe that it is an important point that provides additional motivation for the work we present so we want to leave the suggestion where it is. In fact, we have thought about this issue quite seriously. It is the subject of ongoing theoretical work. We don't want to present this work at this time however, to add weight to our suggestion, we have added a section to the Supplementary Information (Section VIII) that describes the general approach. In summary, the azimuthal spin forces that drive our oscillators are approximately proportional to the square of the volume of the particle (in the Rayleigh regime), while the gradient forces are approximately proportional to the volume. According to the threshold condition (Eq. (3), main text), this means that the threshold pressure, for a fixed optical power, varies $\propto V^{-2/3}$. So, by using nanoparticles, instead of micro-spheres, limit cycles can be formed in ultra-low vacuum. In addition, some sort of cooling protocol is required for further cooling these limit cycles. As mentioned, this is a subject of ongoing research. However, one of our recent articles describes the general principle. In *Cooling the optical-spin driven limit cycle oscillations of a levitated gyroscope*, Arita et al., *arXiv:2204.06925 (2022)*, we show a protocol for cooling limit cycles by parametrically modulating the external forces with a lock in amplifier. Together, these

two steps (reducing the threshold pressure, and parametric force modulation) could allow us to approach the quantum regime. These effects could be further enhanced, perhaps, by scattering into a detuned resonant cavity.

4. (4) I was disappointed to see that despite the detailed theoretical simulations in the supplementary, there are no comparisons (even qualitative) between the experimental and theoretical results in the figures. This could provide validation and potential insight into the relative contributions of the hydrodynamic and optical interactions to the measured data, which is a central theme of the paper. Why have the authors chosen to not show theoretical plots in the main figures?

The reviewer is right to raise this issue. Regrettably, the quantitative connection between experiment and theory was not feasible due to the sensitivity of the experimental system to small defects or imperfections (e.g. optical aberrations of trapping beams). We have revised the manuscript to clarify this issue. In particular, we have added a sub-section to the main text (Section IID) and added a section to the the Supplementary Information (Section VI). The issue is summarized in the main text, and described in more detail in the SI.

Before describing these revisions, we make the following points in defense of our work:

- The main claim of our article is that mesoscopic, light driven limit cycle oscillators can be synchronized in the underdamped regime. Our work is motivated in two ways, (i) cooling such a system towards the quantum regime would give experimental access to mesoscopic quantum dynamic effects, (ii) this work could form a starting point for the exploration of non-Hermitian or topological effects (e.g. the topological skin effect) or even time crystals, in this regime, with clear applications to sensing. We believe that the main claim of the paper is robust and supported by the experiments, theory and simulations. All reviewers appear to agree, with some concerns voiced about the potential for cooling, which we address above.
- We have good qualitative agreement between experiment, theory and simulation - which supports our claims.
- The experimental system is intrinsically sensitive, making it difficult to get quantitative agreement between experiment and theory. In particular, the non-conservative optical forces responsible for the observed phenomena are very sensitive to optical aberrations. Although this can be seen as a weakness, it could be also interpreted as an indication of the robustness of mechanisms - even when non-conservative driving forces vary greatly, synchronization can still be obtained in this system by varying the optical power.

To clarify the qualitative relationship between experiment and theory we have:

- (a) We have revised Fig. 1, adding some simulation results showing the formation of synchronization emerging from the CoM mode. These simulations are described in the revised figure caption, and in the accompanying text. We note the difficulty in quantitatively comparing experiment and simulation by adding following text:

We note that the threshold power is very sensitive to any imperfections in the system. Our numerical stochastic simulation, Fig. 1(d) assume a perfect system without beam

misalignment, aberrations or asymmetries. Under these circumstances, the value of threshold power is (for the experimental value of pressure) an order of magnitude smaller than that observed in our experiments. This has a significant consequence for particle-particle interaction and results in relatively stronger hydrodynamic interactions which favour formation of the CoM QM. Quantitative comparison between theory and experiments was not feasible, see Fig. 1(d). A more detailed discussion is provided in section II D.

(b) Added a subsection (Section II D) to the main text, reading as follows:

The treatment given above, for the sub-threshold behaviour of our spin-driven oscillators, provides sound qualitative insight into the behaviour observed in the experiment. However, the physical system is intrinsically sensitive to small departures from ideality and this makes a direct, quantitative comparison difficult to make in this case. The causes of this sensitivity are described in detail in Supplementary Note (VI). To summarise, consider the threshold condition for a single spin-driven oscillator. The complex eigenvalue describing the linear motion is $\lambda = Kr + iK\phi$ [31] so that the threshold power is $P = 2Kr/mK\phi^2$, with the usual Stokes drag. The azimuthal stiffness, $K\phi$, derives from inhomogeneity in the circularly polarized field [31] and is acutely sensitive to aberrations or other beam deformities. These aberrations may be slightly different for each beam, inducing small asymmetries which are absent in the idealized model described above and are responsible for the weak coupling observed between the Br and CoM QMs.

In addition, η is subject to some experimental uncertainty. This parameter is evaluated from measurements of the pressure out of the relatively small (5 mm inner size) vacuum chamber, which may differ from the pressure inside the chamber, and we assume a kinetic relationship between the pressure and bulk viscosity [55]. Laser heating of the particle may induce additional changes in the effective value of η [60]. In combination, these factors contribute to a broad uncertainty in the threshold power for a given pressure. In the model, which assumes perfect, aberration free Gaussian beams, we have a threshold powers of ≈ 20 mW, at the experimental pressure of 17 mbar, varying slightly with separation (due to interactions). This should be compared with the experimental values of ≈ 180 mW at the same pressure. The size of this discrepancy is due to the intrinsic sensitivity of the system. Although we use wavefront correction methods to reduce aberrations, small (≈ 8 - 10 %) reflections from the vacuum chamber windows are unavoidable and perturb the beam, as is confirmed by the deformations in the particle trajectories. These considerations effect the relative influence of optical and hydrodynamic coupling in the model compared with experiment. However, we note that calculations and simulations performed at a wide range of pressures (≈ 15 - 100 mbar and a commensurate range of threshold powers) reproduce the qualitative features observed in the experiment, and predicted by the preceding analysis. That is, we observe a power dependence, below threshold, that favours one or other of the QMs, followed by a transition giving way to synchronized motion. At steady state, azimuthal optical and drag forces are always balanced, irrespective of the threshold conditions. However, relative influence of these forces in radial directions changes greatly, shifting in favour of optical coupling in the experiment (due to the relatively higher power). In general, the CoM mode is favoured in simulations, while the Br mode is more likely to arise in the experiment. The occurrence of Br or CoM in the experiment is dependent on beam

separation, d , confirming the role of coupled optical forces.

- (c) Added a section to the Supplementary Information (Section VI), in which we quantify the effect of deformations of the optical beam and other experimental parameters. In particular, we plot the threshold power as a function of a particular beam distortion (ellipticity), showing how it can vary by an order of magnitude for relatively weak deformations. We also discuss uncertainty in the Stokes' drag.
5. (5) The Discussion section emphasises that hydrodynamic interactions play a key role in synchronisation. This may be true, but there is no convincing evidence for this in the experimental data. In fact, every mention of hydrodynamic interactions in the main text is in conjugation with optical interactions, with no attempt to isolate hydrodynamic contributions. The theoretical simulations in the supplementary indicate that such interactions play an important role (and might even be sufficient) for synchronization, but without a connection to the main experimental figures (see comment #4), I do not think these theoretical results are sufficient to warrant the strong claim in the Discussion session about the role of hydrodynamic interactions.

There are two issues here. The first is the degree of certainty with which we assert that hydrodynamic coupling *plays a key role*. The second is our evidence for making this assertion. The answer to the first issue is that it was not our intention to express absolute certainty. Rather than *emphasising the key role*, the submitted article reads as follows:

In addition, our results suggest that hydrodynamic interactions play a role in the formation of coordinated motion in both the linear and non-linear regimes

i.e. it is a suggestion, and we do not say that it is *key*. The suggestion is based on evidence outlined in the SI. To soften this assertion yet further, the text has been amended as follows:

In addition, our results suggest that hydrodynamic interactions could play a role in the formation of coordinated motion in both the linear and non-linear regimes.

The reviewer goes on to say that our theoretical results are insufficient to justify a strong claim. We agree - but we did not think our claim was strong and we hope now that it is less strong.

Despite the fact that we (hopefully) do not make a strong claim on this issue, our assertion is based on some evidence and reason:

- In a previous article ([60] in the main text of the paper under review, or *Stochastic Dynamics Of Optically Bound Matter Levitated In Vacuum*, Svak et al, Optica (2021)), we measured the hydrodynamic coupling between spheres in a similar environment. In this previous article, we used linearly polarized light and the optical forces were effectively conservative. We measured the variation in the effective drag between breathing and centre of mass *normal* modes, finding that the Oseen tensor provides a qualitative guide to the interaction. If anything, our results showed that the Oseen tensor underestimates hydrodynamic coupling in this regime. This makes it easier to resolve the influence of hydrodynamic coupling. In contrast, in non-conservative systems (like the one studied in the manuscript under review) dissipative coupling and non-conservative

optical coupling are intermingled, making it difficult to separate the effects. This can be easily seen in the linear (sub-threshold) regime where the characteristic frequency, Eq. (2), has an imaginary part that combines non-conservative optical forces and hydrodynamic damping.

- In a non-conservative system (like the one studied in the manuscript under review), steady state is attained when energy entering the system (from the non-conservative forces) balances that dissipated (which is primarily hydrodynamic, although non-conservative forces can also damp motion - see 'Stochastic Hopf Bifurcations' by Simpson et. al. PRE (2021)). Because of this balance, non-conservative azimuthal forces and hydrodynamic forces are similar in size. If hydrodynamic interactions exist - and are appreciable in size compared with the direct Stokes drag - they must influence the coordinated motions of our particles. Our previous measurement ([60]), described above, demonstrate the influence of such a coupling. It is therefore reasonable to *suggest* that it plays a role in the system studied in the article under review. In fact, it is hard to imagine (given the balance between azimuthal optical and hydrodynamic forces) that it plays no role at all.

Lastly, we comment on the reviewers (completely valid) remark that, **In fact, every mention of hydrodynamic interactions in the main text is in conjugation with optical interactions, with no attempt to isolate hydrodynamic contributions.** As follows, there are several reasons why he not attempted to separate hydrodynamic and optical interactions in this paper:

- (a) The first reason is that this is a hard problem! and, we suggest, beyond the scope of this paper which is already long and complex (or, *deceptively simple*) to use the phrase coined by another reviewer. The reason for this difficulty is that, as the reviewer correctly summarizes, optical and hydrodynamic interactions always occur in conjugation. As described above, this is a necessary characteristic of a non-conservative system at steady state, in which the work performed by non-conservative and hydrodynamic forces must balance, as described above. Since they balance, these interactions do not vary independently - making them hard to separate. We note that, in the low Re systems, no particular attempt is made to separate non-conservative forces and dissipative coupling either. In experiments involving hydrodynamic coupling between spheres in static optical traps, the forces are known (e.g. from calibrating the optical traps), and the role of hydrodynamic interaction is inferred (in addition, optical coupling forces are uniformly neglected in these systems, which may be a questionable assumption). Meanwhile, in experiments on low Re hydrodynamic synchronization, driving force profiles are set by the experimenter so that the only remaining unknown is the hydrodynamic interaction.
- (b) The second reason is that it seems almost (but not quite) seems superfluous. In the final point below, we describe a sophisticated scheme, developed in our group, that *could* separate optical and hydrodynamic interactions. However, it is far simpler to consider a system in which hydrodynamic and optical coupling are automatically separated i.e a conservative system. This can be simply appreciated by considering the characteristic frequency and autocorrelation function for a linear system Eqns. (2) and (4) in the main text. For conservative systems the eigenvalue (λ) is purely real - so the imaginary part of the characteristic frequency, Eq. (2) is purely hydrodynamic as is the decay of the autocorrelation (Eq. (4)). Differences in the decay lengths for the autocorrelations of different *normal* modes are due to hydrodynamic coupling, the nature of which depends on the normal mode. We have previously done such measurements, see [6].

- (c) There are more rigorous ways to distinguish optical and hydrodynamic coupling in these complex optomechanical systems. One such method has been developed in our group, see *Bayesian Estimation of Experimental Parameters in Stochastic Inertial Systems: Theory, Simulations, and Experiments with Objects Levitated in Vacuum*, S'iler et al., arXiv:2212.14043. It relies on Bayesian inference. In principle, it could be applied to the trajectories measured in these experiments. However, this requires considerably more work to adapt the method for this application and to ensure the validity of the results it provides. Although we certainly intend to undertake such work in the future, we genuinely feel that this is beyond the scope of the current article. Inference of hydrodynamic interactions in non-conservative, underdamped systems is challenging, and worthy of an article in its own right.

In summary, we agree with the reviewer - separating optical and hydrodynamic interactions is a worthy endeavour, and one we are currently considering. However, we feel that it is a significant challenge, and one we worthy of a distinct article.

6. (6) I appreciate the care taken by the authors to calculate the Shannon entropy as an estimate of the synchronization strength. However, this inevitably raises questions in the reader about why, even well above threshold, the Shannon entropy is ≈ 0.4 , which suggests that the synchronization is still stochastic.

The reviewer is correct, the thermodynamic fluctuations are certainly apparent in the synchronized state. Since the work performed by non-conservative and viscous forces must balance at steady state - dissipative forces must always be significant and, as a consequences so must thermal fluctuations. Synchronization is a generic effect and we should expect it to emerge, whenever limit cycles interact. In the mesoscopic regime, the main question is whether or not the system can access mechanisms that are strong enough to overcome the fluctuations. We feel that our results conclusively demonstrate that the answer in this (quite extreme) regime is affirmative. No changes to the paper have been made.

7. (7) Can the authors provide possible mechanisms for the "shift to slightly greater phase differences" in reference Fig 4c?

It looks as if the shift occurs in the same direction as the preferential phase shifts, suggesting that the imbalance in the systematic driving forces, connected with detuning, is responsible for the shift. We have added the following sentence to the main text:

We note that this broadening is in favour of the faster oscillator, and may reflect the growing difference in the driving forces connected with detuning.

A. Minor Comments

Minor comments from reviewer #2 are at the level of typo's. We thank the reviewer for calling our attention to these issues, each of which has been corrected in the main text as follows.

- 8) Fig 1a caption: Typo "fromve".

Typo corrected

- 9) Fig 1d caption: "further developed stable" and "further grows" are hard to understand. Typo corrected

- 0) Sec 2a text, paragraph 2: Two references to Fig 1f, which does not exist. Corrected, with revised Fig. 1.

- 11) Sec 2c text, first paragraph on page 6: "in for" should be "for".
Typo corrected.

- 12) Sec 2D, brackets not closed in references in supplementary.
Typo corrected

- 13) Sec 3 last paragraph: "experiment" should be "experiments"
Typo corrected.

14) The description of what a limit cycle is, given in page 4, needs to appear in the introduction.

The following text has been added to the Introduction immediately after the first reference to limit cycles:

(i.e. stable, self-sustained, preiodic motions)

- 15) Statements that need to be expanded for readability: (a) "Since limit cycles are neutrally stable,...", unclear what "neutral" means, and (b) "a typical noise-induced phase slip", needs a either a reference or a description of the mechanism leading to such phase slips, especially because they become relevant later in Fig 4.

The following text has been added to the section II E, after the reference to neutral stability:

(that is, in a single oscillator, each particle is equally stable at any point on the limit cycle).

The following sentence has been inserted after the mention of *phase slips*:

This happens when fluctuations push the oscillators far out of the synchronized state and, instead of reversing back into it, synchrony is restored after one of the oscillators when the phase of one oscillator increases by 2π relative to the other [61].

Where [61] is the classic text by Pikovsky, which contains a detailed description of the effect, albeit in a generic oscillator.

III. REVIEWER #3

We thank the reviewer for their thoughtful comments. Below, we give a point-by-point response. Some of these comments are also made by reviewers #1 and #2, in which case we refer back to the responses given above.

A. Scientific Comments

- 1. The ambient pressure at which the experiments take place (17 mbar) is quite high for typical levitated optomechanics experiments, particularly those that have any interest in approaching the quantum regime (which typically utilize pressures 7 to 10 orders of magnitude lower). At 17 mbar, it seems that any interesting quantum state would decohere on extremely short timescales (likely much shorter than any other timescale in the reported experiments), but the possibility of using "cooling protocols" to push these experiments into the quantum regime is suggested (although primarily only in the Introduction and Discussion sections). Is there any plausible path to quantum behavior from these experiments? If it would require more than cooling, that should be noted.

We thank the reviewer for calling our attention to this issue. Similar concerns were expressed by reviewer #2, point 3 (see above). In summary our response is as follows. First - we mention, only briefly, the possibility of observing quantum effects. We hope that our comment provides motivation for the work we present, it is not part of the work itself. It is one application amongst several that we suggest. Second - we believe that our suggestion is credible, if challenging. In the main text we have added a comment in which we refer to a new section in the Supplementary Information (see Section VIII) which sketches a possible route. The approach consists of two steps. First we shift to nanoparticles. This reduces the size of the non-conservative azimuthal forces relative to the gradient forces. The threshold pressure (below which limit cycles form) is $\propto V^{2/3}$, allowing us to operate at very low pressures. Next we apply parametric modulation, driven by a lock in amplifier. This step is described in our previous paper *Cooling the optical-spin driven limit cycle oscillations of a levitated gyroscope*, Arita et al., *arXiv:2204.06925 (2022)*, which has recently been accepted for publication in Communications Physics. Some beam shaping with the SLM may be required to reproduce the effects observed in our Commun. Phys. paper.

- 2. The ambient pressure is only stated in one place (in the figure 1 caption), but this seems to be a critical parameter of the system based on the analysis in the supplemental information.

How sensitive are the experiments to the ambient pressure? Was similar behavior observed over some pressure range (although presumably with different optical powers), or does the pressure need to be carefully tuned?

We thank the reviewer for raising this important point. A similar point was also raised by reviewer #2, point 4 (see above). We have added a new subsection to the main text, section II D, which describes the system sensitivity and the connection between theory and experiment. This subsection refers to a more detailed account in the Supplementary Information (section XX). In summary, the system is very sensitive to small beam aberrations. These degrade the (small but crucial) non-conservative azimuthal forces. This process has a strong effect on the threshold power (see Eq. (3) in the main text) that is difficult to control in the experiment. Finally, we clarify the reviewers comments regarding pressure and power. The threshold condition, Eq. (3) in the main text, is a condition for the stability of the fixed trapping point. It specifies a value of the ratio, P/γ^2 below which the fixed point is stable and above which it is unstable. When the fixed point destabilizes, the system undergoes a Hopf bifurcation and (dynamically) stable limit cycles form. This picture is complicated by thermal fluctuations - i.e. both the fixed point and the limit cycles are noisy. The threshold condition can be crossed either by varying the power, P , or the effective drag, γ , which is proportional to pressure in this regime. The dynamics do not depend qualitatively on whether P or γ are varied - however, the fluctuations about the dynamical attractors (fixed point or limit cycle) do depend on the relative size of the power and drag. Thus, similar effects could be observed by varying the pressure and holding the power constant. We have presented result showing the formation of limit cycles induced by pressure variation in a recent article, mentioned above (*Cooling the optical-spin driven limit cycle oscillations of a levitated gyroscope, Arita et al., arXiv:2204.06925 (2022)*). To clarify this final point we have inserted the following text at the end of section IIB in the main text.

As shown in [52] for single oscillators, similar transitions can be induced by variations in pressure, rather than the power.

- where [52] is the paper referred to immediately above.

- 3. In the analysis of the effects of the ambient pressure in the supplemental information, it appears that the only effect considered is drag (including coupling between the particles due to drag). Were radiometric forces (due to the interaction of thermal gradients and gas) considered? Is there justification for ignoring these forces? Anecdotal evidence suggests that radiometric forces may be very important at these size scales and at high pressure (near atmospheric pressure). A quick estimate suggests a mean free path of gas molecules of 4 micrometers at 17 mbar and room temperature, quite close to the diameter of the particles (1 micrometer) and thus in the regime where radiometric forces could be significant. It is still quite plausible that the speed of the particles is so high that drag forces are dominant, but I did not see any discussion of this in the manuscript.

We do not believe that radiometric forces contribute greatly to the results we observe here. As the reviewer suggests, inertial effects connected with the high deterministic velocities dominate in this regime. In addition, as described below, the effect does not appear in our other experiments conducted in this regime and, in addition, radiometric forces would probably be suppressed in our system due to rapid spinning of the particle caused by the same material absorption that would be required for radiometric forces to be appreciable.

Below we provide some discussion of the issue.

In the following paragraphs we consider the possible influence of radiometric forces. As will become clear, it is difficult to rigorously evaluate these forces, for this system. Nevertheless, for the reasons we provide, we believe that their impact is either negligible or minor and quantitative. We are convinced that radiometric forces are certainly not responsible for the qualitative results we report (namely stochastic orbital motion, instability, limit cycle formation and synchronization), although it is possible that they have a minor quantitative effect.

First, we start with some empirical observations. We note that the qualitative effects we report depend critically on circular polarization (whereas, of course, radiometric forces do not). They are completely absent for linearly polarized beams. This can be seen in our previous experiments, performed on a single spin oscillator, under similar ambient conditions (Svak et al., Nat. Comms. (2018)). Furthermore, the fluctuations in the position of the micro-sphere satisfy equipartition, when linearly polarized light is used. Together, these two experimental observations indicate that the effects, we observe in the paper under review, are not caused by radiometric forces and, in addition, that the radiometric forces is small in this regime.

Next, we make some qualitative comments about how radiometric forces might behave in our system. As mentioned above, we will show that these forces are hard to evaluate in our system and that even if they arose, other mechanisms would act in such a way as to suppress their effect. As the reviewer doubtless appreciates, radiometric forces are connected with temperature gradients, which give rise to induced flows in the ambient gas, thereby generating forces. These temperature gradients may either be (i) externally imposed, or (ii) arise due to asymmetric heating of the particle(s). We discard (i) immediately, since externally imposed temperature gradients in a rarified gas are necessarily low and would, in any case, just provide a small constant force offset. Moving to possibility (ii), asymmetric heating requires material absorption (usually thought to be negligible for polystyrene spheres at these wavelengths). We hypothesise (contrary to the experimental evidence given above) that material absorption is present and strong enough to induce a non-negligible radiometric force. Now, at a particular instant, heating is likely to be asymmetric, due to the orbital motion (which is stochastic below threshold, deterministic above threshold). In principle, this would preferentially warm regions of the particle that are closer to the beam axis, where the optical intensity is greatest. In turn, this would generate radial radiometric forces which could not cause limit cycle formation, but could influence the path of the limit cycle. However, the physical situation is more complicated. The same absorption that is required for heating would also enable coupling with the spin optical angular momentum of the beam, causing the particle to rotate (spin) about an axis through its centre. Unless the particle spins at the same rate as it orbits (which is not possible for all optical powers, see our recent article, Arita, Yoshihiko, et al. "Cooling the optical-spin driven limit cycle oscillations of a levitated gyroscope." arXiv preprint arXiv:2204.06925 (2022)), this spinning rotation would cause the temperature distribution within the particle, and on its surface, to become even and symmetric (like a rotisserie chicken!). This process mitigates the generation of gas flows over the particle surface, and therefore suppresses radiometric forces. It is this combination of spinning, heating and orbiting which makes a rigorous evaluation of the radiometric forces challenging, for this system. Finally, we note that the spinning, orbiting particle would also be subject to forces connected with the Magnus effect. These forces would also act radially, influencing the path of the limit cycle but not causing its existence. However, a back of the

envelope calculation (using the Kutta-Joukowski expression for the lift on a cylinder, and then scaling to the finite size of the sphere) indicates that the Magnus force is completely negligible, even if the particle spins at many GHz - which would be unprecedentedly rapid spinning for a particle of this size.

Finally, it is possible that these mechanisms could induce coupling forces. For example, one hot particle induces local temperature gradients in the gas surrounding the second. In this case, the temperature gradients would be on the order of a/d , where a is the sphere radius and d the distance between the beams. This gives rise to an interaction of the form $\propto 1/d^2$, for large d . Whilst it is difficult to rule this out (except through the empirical observations given above), we note that this form of interaction, a/d , is monotonic. It cannot, therefore, account for the observed switching between CoM and Br quasi modes that are observed in our experiment and theory as the beam separation is varied. This behaviour indicates an interaction that changes sign with varying separation - a well known characteristic of optical interactions.

To summarize, our previous experimental work suggests that radiometric forces are negligible in this regime. In addition, the results we report depend critically on circular polarization and are completely absent for linear polarization - demonstrating that the non-conservative forces are causative here. Finally, the discussion above shows (at least qualitatively) that the absorption required for radiometric forces to appear, would simultaneously cause spinning of the particle, evening out the surface temperature distribution and thereby suppressing radiometric forces. Radiometric coupling may be present, but it is almost certainly weak (on the order of $1/d^2$), and approximately constant.

We have chosen not to discuss these issues in the article since we do not wish to add further complexity to an already complex article, unless it is absolutely necessary.

B. Minor Comments

We thank the reviewer for calling our attention to these minor issues. We have added the experimental parameters to the appropriate section in the main text both in the Methods section and in section II A. We have also corrected the typo!

REVIEWERS' COMMENTS

Reviewer #1 (Remarks to the Author):

After reviewing the revised manuscript, the updated Supplementary Materials and the authors' response to the referees, I conclude that the authors have addressed the issues I raised in the first round of review. This manuscript addresses an interesting model system in which synchronization arises from the interplay of driving, inertia, conservative coupling and nonconservative coupling. The qualitative comparison between experiment and theory suggests that the principal physical principles have been reasonably identified and applied. The system's apparent sensitivity to details of the experimental implementation strikes me as somewhat surprising, but can be left to future studies to sort out. I recommend publication of the revised manuscript.

Reviewer #2 (Remarks to the Author):

I appreciate the detailed responses of the authors. The authors have satisfactorily answered my comments, therefore I recommend publication of the revised manuscript.

However I would request the authors to consider the following, for readability -

The responses have led to two significant additions in the main text - (a) qualitative comparison to theory in Fig 1 and (b) a new section 2D. Adding the theory plots in Fig 1 adds value to the manuscript, although I understand the argument that the simulations provide only a qualitative comparison. However, section 2D is unnecessary in the main text, and the entire part from "To summarise... coupled optical forces" should be in the supplementary Sec 6.

Reviewer #3 (Remarks to the Author):

The revisions have satisfactorily addressed my comments and concerns. I feel the manuscript is much-improved and acceptable for publication.

According to suggestion of Reviewer #2

However, section 2D is unnecessary in the main text, and the entire part from "To summarise... coupled optical forces" should be in the supplementary Sec 6.

The text referred to by the reviewer mostly summarizes the text already in the Supplementary Information (SI). Thus, we have merged the section from the main text with the existing text in the SI so that all necessary information is retained without repetition.